# FairSpace: Search Space Pruning of AutoML for Fairness-Accuracy Trade-off

## Abstract

A major challenge in responsible Machine Learning (ML) engineering is ensuring fairness across multiple protected attributes and their intersections. Existing bias mitigation techniques and Automated Machine Learning (AutoML) systems often fail to address this due to the combinatorial explosion of configurations during hyperparameter optimization (HPO). We propose FAIRSPACE, a fairness-aware framework that jointly performs HPO and dataset-specific feature engineering while strategically pruning the configuration space. FAIRSPACE integrates LLM-assisted feature engineering methods with a bi-objective cost function to balance fairness and accuracy. Experimental results on five widely-used datasets demonstrate that FAIRSPACE achieves win–win outcomes—simultaneously improving fairness and accuracy for 63% of the cases, outperforming state-of-the-art (SOTA) baselines that achieve up to 60%. Moreover, FAIRSPACE achieves these results with approximately 25% less computation time, owing to its targeted pruning strategy as compared to the SOTA AutoML baseline such as FairAutoML. By explicitly tackling intersectional fairness, FAIRSPACE reaches 94% of its outcomes in the *win–win* and *good trade-off* regions, providing a consistent and generalizable foundation for fairness-aware AutoML.

## 1 Introduction

Recent breakthroughs in machine learning (ML) have enabled powerful data-driven solutions across high-stakes decision-making domains such as loan approval, hiring, education, and criminal justice Nilashi et al. (2017); Dahiwade et al. (2019); Appadoo et al. (2020); Mahmoud et al. (2019); Halde (2016); Sheikh et al. (2020); Singh et al. (2021). However, these advancements have also exposed fairness issues Angwin et al. (2016); Biswas & Rajan (2020), with models perpetuating or amplifying historical biases and causing harms to underrepresented groups Friedler et al. (2019); Galhotra et al. (2017); Buolamwini & Gebru (2018). Yet, despite various mitigation techniques proposed in fairness-aware machine learning Kamiran & Calders (2012); Feldman et al. (2015); Hardt et al. (2016), their effectiveness remains inconsistent across datasets, fairness metrics, and deployment contexts Grabowicz et al. (2022); Tizpaz-Niari et al. (2022). Moreover, these methods often fail to adequately address **intersectional bias**, where unfairness arises from combinations of sensitive attributes such as race and gender Chen et al. (2024b).

Recent AutoML-based approaches, such as FairAutoML Nguyen et al. (2023), have shown promise in automating the trade-off between accuracy and fairness through hyperparameter optimization (HPO). By reducing the need for manual configuration and domain-specific expertise, these systems help democratize access to fairness-aware ML Karmaker et al. (2021). Yet, AutoML in its current form is not inherently fairness-aware, and its reliance on large search spaces makes it computationally expensive Weerts et al. (2024). Fairness outcomes depend on both hyperparameters (HP) and feature engineering (FE), making HP-only optimization insufficient Biswas & Rajan (2021). While FairAutoML incorporates limited preprocessing, it still depends on an offline search for promising HPO points, adding overhead and limiting adaptability across datasets. To address these gaps, we introduce FairSpace, a framework designed to eliminate the need for offline search and jointly optimize HP and FE choices, thereby improving efficiency and fairness from the outset. State-of-the-art (SOTA) fairness-aware AutoML still faces three persistent challenges (see Section 2): (1) compounded biases at the intersections of multiple protected attributes are often overlooked, limiting

subgroup fairness; (2) dual-objective HPO substantially enlarges the search space, driving up computational costs; and (3) dataset-specific FE strategies are rarely integrated, resulting in suboptimal preprocessing choices that undermine fairness and accuracy.

We propose FairSpace to overcome these limitations through a two-stage refinement process. First, we incorporate LLM-assisted, dataset-specific feature engineering (FE), where the LLM recommends preprocessing strategies (e.g., imputation, encoding, normalization, selection) tailored to each dataset. These recommendations determine which preprocessing methods to include or exclude, making the configuration space dataset-aware. Second, we prune the hyperparameter (HP) search space using a guided exploration strategy. The HP space is divided into batches, and only those showing distinct performance (e.g., accuracy) are explored further. Batches that achieve a satisfactory fairness–accuracy trade-off, measured with the Fairea methodology Hort et al. (2021), are preserved along with their neighbors. Within these promising regions, focused exploration identifies the most effective solutions, while the rest are pruned, reducing search cost without sacrificing fairness-awareness. Finally, models are trained on this refined space using a bi-objective cost function that minimizes loss while jointly accounting for fairness and accuracy. In doing so, our work contributes to responsible ML engineering, an emerging area within SE that emphasizes trustworthy and scalable ML pipelines.

We implemented FairSpace on top of the open-source AutoML package Auto-Sklearn Feurer et al. (2020). We evaluated FairSpace on five datasets Aggarwal et al. (2019); Galhotra et al. (2017); Udeshi et al. (2018) from recent fairness literature Biswas & Rajan (2020); Nguyen et al. (2023); Chen et al. (2022); Xiao et al. (2024). We compared FairSpace against six SOTA baselines, covering each stage in fairness optimization (i.e., pre-processing, in-processing, and post-processing). FairSpace consistently achieves "win–win" outcomes—improving both fairness and accuracy—in 63% of cases, outperforming both the best prior AutoML-based method (FairAutoML, 38%) and the strongest non-AutoML method (MirrorFair, 60%). Moreover, FairSpace achieves an average 94% reduction in bias, with improvements extending to intersectional fairness metrics. It also reduces runtime by about $\sim 25\%$ compared to baselines, owing to its pruning of the hyperparameter search space. We make the following **contributions**:

1. **Fairness-Aware HPO:** We present *FairSpace*, an HPO and FE framework that jointly optimizes accuracy and fairness by exploring the configuration space with fairness-aware search strategies, thereby mitigating bias while maintaining predictive performance.

2. **Support for Multiple and Intersectional Protected Attributes:** FairSpace supports bias mitigation for both single and multiple protected attributes, integrating intersectional metrics throughout search and evaluation.

3. **LLM-Assisted Feature Engineering:** FairSpace leverages LLMs to suggest context-aware preprocessing strategies, filtering out irrelevant options to shrink the search space and enhance fairness-awareness.

4. **Open-Source Release:** We build FairSpace on Auto-Sklearn and release code, models, and results to ensure reproducibility and advance fairness-aware AutoML Anonymous (2025).

## 2 Motivation

The Hyperparameter Optimization (HPO) of an ML model Wu et al. (2019); Yang & Shami (2020); Liao et al. (2022) has shown promise in automatically identifying optimal fairness–accuracy trade-offs Nguyen et al. (2023); Chen et al. (2022). However, moving from single-objective (accuracy) to dual-objective (fairness + accuracy) optimization, as illustrated in Figure 1 for RandomForest, significantly enlarges the hyperparameter space and increases computational cost, since more configurations need to be explored. These challenges are particularly pressing in the context of Software Engineering for ML pipelines, where generalization, efficiency, and fairness must be jointly addressed.

Beyond hyperparameter configurations, the challenge of fairness-aware modeling is further compounded by feature engineering, which shapes not only predictive performance but also fairness outcomes. Prior work has

demonstrated that different feature engineering methods can mitigate bias depending on dataset characteristics Biswas & Rajan (2021). For instance, one-hot encoding can inflate dimensionality and disproportionately affect minority groups linked to protected attributes Mougan et al. (2023). Dataset meta-features—such as attribute types, distributions, and correlations—determine whether a method improves performance and fairness or instead introduces unintended bias Yab et al. (2022). For example, encoding strategies affect categorical attributes differently than numerical ones Mougan et al. (2023), while normalization or scaling techniques become critical when feature ranges vary widely. As a result, the selection of feature preprocessing and transformation methods (e.g., encoding, scaling, imputation, or feature selection) must be carefully aligned with dataset properties. Although these two phases are deeply intertwined, as feature engineering choices directly influence the effectiveness of HPO, they are often treated in isolation Salazar et al. (2021), resulting in suboptimal and inefficient searches for fair models.

In addition, AutoML fairness methods like FairAutoML can balance fairness and accuracy when considering a single sensitive attribute. However, they fall short in capturing intersectional fairness, where multiple attributes (e.g., race and gender) combine to form subgroups. Since FairAutoML evaluates one attribute at a time and uses group fairness metrics that ignore these overlaps, it often overlooks biases at intersections. As a result, FairAutoML fails to protect the most vulnerable subgroups who experience compounded unfairness.

To address these challenges, FairSpace integrates feature engineering with HPO, leveraging LLMs to generate dataset-specific methods from meta-feature analysis. This filters out irrelevant preprocessing and steers the search toward configurations that better balance fairness and accuracy. By coupling LLM-guided feature engineering with fairness-aware HPO, FairSpace adapts the configuration space to the dataset, avoiding redundant exploration while ensuring preprocessing and modeling choices remain aligned with data properties. This unified approach enables equitable yet accurate outcomes across both single and intersectional protected attributes, aiming to address the limitations of prior methods.

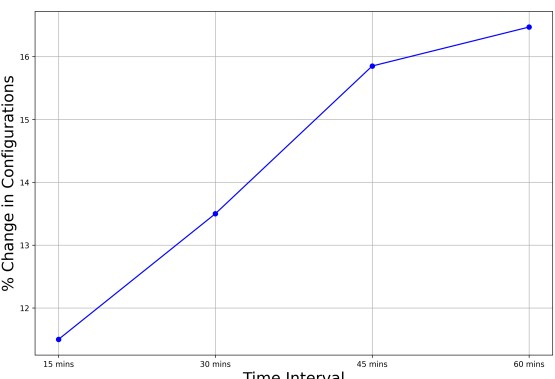

Figure 1: Average percentage increase in configurations when transitioning from single-objective to bi-objective optimization across datasets.

## 3 Background

**Fairness-accuracy trade-off.** Achieving improvements in both fairness and accuracy is a well-known challenge in ML bias mitigation, as efforts to increase fairness often come at the cost of predictive performance. To rigorously evaluate this trade-off, we utilize the Fairea framework Hort et al. (2021), which offers a systematic approach for evaluating the trade-off between fairness and accuracy.

Fairea visualizes each method's results on a two-dimensional plane—fairness versus accuracy—with the baseline defined by the original and mitigation model's coordinates. Fairea applies controlled mutations to model predictions, randomly altering subsets of outputs at various intensities. Each outcome is then classified into one

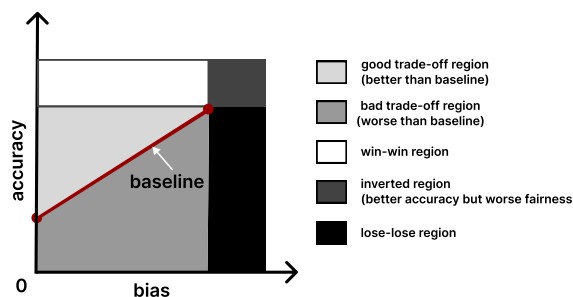

Figure 2: Bias-Accuracy Trade-Off in Fairea Hort et al. (2021).

of five regions: *lose-lose* (both accuracy and fairness decrease), *bad trade-off* (performance is worse than the baseline), *inverted* (improvement in accuracy but a drop in fairness), *good trade-off* (better performance than the baseline), and *win-win* (simultaneous improvement in both fairness and accuracy) as shown in Figure 2. This classification shows how well a method performs compared to the original model and highlights

the challenge of achieving win–win outcomes, stressing the need for careful design to balance fairness and accuracy. While Fairea offers the evaluation of the trade-off, we further leverage multiple established fairness metrics with Fairea, including group and intersectional fairness.

**Group and Intersectional Fairness.** A central aim of fair machine learning is to prevent systematic advantage or disadvantage based on protected attributes such as sex, race, age, or disability Bellamy et al. (2018); Binns (2018); Friedler et al. (2019). **Group fairness** seeks similar outcomes across privileged and unprivileged groups, and is commonly quantified by well-established metrics suitable for empirical comparison and real-world use Chakraborty et al. (2021); Chen et al. (2022); Nguyen et al. (2023); Tizpaz-Niari et al. (2022); Xiao et al. (2024). However, group-level analysis alone may overlook unfairness that arises when multiple attributes intersect. **Intersectional fairness** addresses this by examining subgroups defined by combinations of protected attributes, thus capturing compounded disadvantages that may be hidden when attributes are considered separately Chen et al. (2024b); Ghosh et al. (2021); Subramanian et al. (2021). To capture fairness across all subgroups defined by multiple protected attributes, we use worst-case and average-case variants. Let $\mathcal{G}$ denote the set of all intersectional groups (e.g., race $\times$ gender) Chen et al. (2024a). To quantify these notions, we use the following metrics (all reported in absolute value so that lower is fairer) Chakraborty et al. (2020); Hort et al. (2021):

$$\text{SPD} = \Pr[\hat{y} = 1 \mid z = 0] - \Pr[\hat{y} = 1 \mid z = 1]$$

$$\text{EOD} = \Pr[\hat{y} = 1 \mid y = 1,\, z = 0] - \Pr[\hat{y} = 1 \mid y = 1,\, z = 1]$$

$$\text{AOD} = \tfrac{1}{2}\Big(\big|\Pr[\hat{y} = 1 \mid y = 0, z = 0] - \Pr[\hat{y} = 1 \mid y = 0, z = 1]\big|$$
$$+ \big|\Pr[\hat{y} = 1 \mid y = 1, z = 0] - \Pr[\hat{y} = 1 \mid y = 1, z = 1]\big|\Big)$$

**Worst-case:**

$$\text{WC-SPD} = \max_{g_1, g_2 \in \mathcal{G}} |\Pr(\hat{Y} = 1 \mid g_1) - \Pr(\hat{Y} = 1 \mid g_2)|$$

$$\text{WC-EOD} = \max_{g_1, g_2 \in \mathcal{G}} |\Pr(\hat{Y} = 1 \mid Y = 1, g_1) - \Pr(\hat{Y} = 1 \mid Y = 1, g_2)|$$

$$\text{WC-AOD} = \max_{g_1, g_2 \in \mathcal{G}} \tfrac{1}{2}\Big(|\Pr(\hat{Y} = 1 \mid Y = 1, g_1) - \Pr(\hat{Y} = 1 \mid Y = 1, g_2)|$$
$$+ |\Pr(\hat{Y} = 1 \mid Y = 0, g_1) - \Pr(\hat{Y} = 1 \mid Y = 0, g_2)|\Big)$$

**Average-case:**

$$\text{AC-SPD} = \tfrac{1}{|\mathcal{G}|} \sum_{g \in \mathcal{G}} |\Pr(\hat{Y} = 1 \mid g) - \Pr(\hat{Y} = 1)|$$

$$\text{AC-EOD} = \tfrac{1}{|\mathcal{G}|} \sum_{g \in \mathcal{G}} |\Pr(\hat{Y} = 1 \mid Y = 1, g) - \Pr(\hat{Y} = 1 \mid Y = 1)|$$

$$\text{AC-AOD} = \tfrac{1}{|\mathcal{G}|} \sum_{g \in \mathcal{G}} \tfrac{1}{2}\Big(|\Pr(\hat{Y} = 1 \mid Y = 1, g) - \Pr(\hat{Y} = 1 \mid Y = 1)|$$
$$+ |\Pr(\hat{Y} = 1 \mid Y = 0, g) - \Pr(\hat{Y} = 1 \mid Y = 0)|\Big)$$

*Group Fairness Metrics:* Here, $\hat{y}$ denotes model prediction, $y$ is the true label, and $z$ is the binary sensitive attribute Nguyen et al. (2023); Chen et al. (2022); Xiao et al. (2024).

*Intersectional Fairness Metrics:* To capture fairness across all subgroups defined by multiple protected attributes, we use worst-case and average-case variants. Let $\mathcal{G}$ denote the set of all intersectional groups (e.g., race $\times$ gender) Chen et al. (2024a), where each element $g \in \mathcal{G}$ represents one such subgroup (e.g., Black women, White men). In the definitions below, $g$, $g_1$, and $g_2$ refer to individual intersectional groups drawn from $\mathcal{G}$.

## 4 FairSpace: Fairness Aware HPO

The FairSpace approach operates within an AutoML pipeline, taking a dataset with protected attributes and a machine learning model as input to produce models that balance fairness and accuracy. FairSpace proceeds in four steps. First, it initializes the hyperparameter (HP) search space by enumerating candidate configurations and, using dataset meta-features, obtains LLM-assisted recommendations for preprocessing and feature-engineering methods, along with suggested parameter ranges. These recommendations narrow the HP space. Second, it conducts a guided exploration across the combinations to evaluate their fitness in terms of fairness and accuracy. Third, it prunes combinations that do not achieve a satisfactory trade-off. Finally, it trains the model on the pruned HP space together with the selected feature-engineering methods, using a bi-objective cost function that jointly optimizes accuracy and the chosen fairness metric.

Table 1: ML algorithms and hyperparameter domains.

| Algorithm | Number of HPs | Hyperparameter Domains/Values |
|---|---|---|
| Logistic Regression | 5 | penalty $\in$ {l1, l2, elasticnet, None}
C $\in$ $\{10^{-5}, 10^{-4}, 10^{-3}, 0.01, 0.1, 0.5, 1, 5, 10, 100\}$
dual $\in$ {T,F}
fit_intercept $\in$ {T,F}
solver $\in$ {liblinear, saga, sag, lbfgs, newton-cg, newton-cholesky} |
| Support Vector Machine | 6 | C $\in$ $\{0.01, 0.1, 1, 10, 100, 1000\}$
kernel $\in$ {linear, rbf, poly, sigmoid}
gamma $\in$ {scale, auto}
degree $\in$ $[2, 9]$
coef0 $\in$ $[0.0, 1.0]$
shrinking $\in$ {T,F} |
| Multi-Layer Perceptron | 7 | hidden_layer_sizes $\in$ {(50,), (100,), (150,), (100,50), (100,100), (128,64,32), (64,64)}
activation $\in$ {relu, tanh, logistic, identity}
alpha $\in$ $\{10^{-5}, 10^{-4}, 10^{-3}, 0.01, 0.1\}$
learning_rate_init $\in$ $\{0.0001, 0.001, 0.01, 0.1\}$
solver $\in$ {adam, sgd}
learning_rate $\in$ {constant, invscaling, adaptive}
early_stopping $\in$ {T,F} |
| Random Forest | 6 | n_estimators $\in$ $[100, 1000]$
criterion $\in$ {gini, entropy}
max_features $\in$ $[0.1, 0.9]$
min_samples_split $\in$ $[2, 20]$
min_samples_leaf $\in$ $[2, 20]$
bootstrap $\in$ {T,F} |

### 4.1 Initialize Search Space as Batches of HP Configurations

We define a broad yet systematic range of hyperparameter (HP) configurations. First, we establish an initial set of HPs along with their respective configuration domains. The choice of values for each HP depends on its type and its relationship to model performance. For parameters that influence performance in a near-linear manner (e.g., *n_estimators* in RandomForest), we use evenly spaced intervals across the range. In contrast, for parameters that exhibit non-linear effects (e.g., *learning_rate*), we adopt a logarithmic scale to ensure finer granularity in regions of higher sensitivity. These design choices align with best practices in hyperparameter optimization and search space construction, as established in prior work Bergstra & Bengio (2012); Schratz et al. (2019).

To generate the full range of HP combinations, we construct the Cartesian product of all selected values, ensuring complete coverage of the search space. A verification function filters out invalid configurations (e.g., inconsistencies between Logistic Regression and SVM), and only validated configurations are passed to evaluation. In total, the search space spans 24 HPs across four algorithms, as shown in Table 1. This design provides sufficient diversity to evaluate FairSpace's ability to prune large configuration spaces while maintaining accuracy–fairness trade-offs.

### 4.2 LLM-Assisted Component Selection

We use prompting to guide large language models in selecting dataset-specific preprocessing and feature engineering methods for FairSpace. The motivation stems from the fact that traditional AutoML pipelines include a vast number of preprocessing techniques, many of which directly affect fairness depending on dataset properties and the classifier used. We instantiate LLMs as meta-guides that map dataset meta-features to preprocessing configurations. This approach reduces computational overhead while adapting the pipeline more effectively to the dataset.

The overall prompting workflow is illustrated in Figure 3, beginning with dataset and classifier selection, followed by prompt construction in DSPy Khattab et al. (2024). The meta-features considered include the representation of sensitive groups, class imbalance, under-representation of minority groups, the distribution of missing values across groups, correlations between features and sensitive attributes, overall dataset size, and the distributional shape of features. In practice, these meta-features are provided as structured inputs to DSPy's declarative prompting interface, which guides the LLM in generating preprocessing and transfor-

mation recommendations. This ensures that prompts are consistent, modular, and reproducible, rather than relying on ad-hoc natural language descriptions.

## Prompt 1

Here are the meta-features of the <Dataset Name>: <Meta Features>. The goal is to mitigate the bias in the <Classifier> model.

Based on these meta-features what <Data Transformation Categories> should be applied?

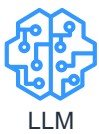

LLM

## LLM Response

Thank you for providing the dataset meta-features. Based on these characteristics, I recommend:

**<Data Transformation Stages>**

## Prompt 2

Based on the meta-features and the <Data Transformation Stage> method, recommend specify the exact parameters required for that method in JSON format.

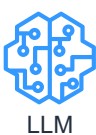

LLM

## LLM Response

Here are the parameter of the <Data Transformation Stage> method in the JSON format.

**<Data Transformation Stage(*params)>**

Figure 3: LLM Prompting Strategy for the selection of feature engineering methods.

To ensure consistency and reproducibility, we follow structured prompt design principles. As shown in Table 2, preprocessing methods are organized into transformation categories, each linked to specific feature input types and their expected outputs after prompting. Each prompt includes the dataset's meta-features, the protected attribute, and the dual objective of accuracy and fairness. The instruction style is directive, requiring the model to recommend from a fixed set of transformation categories rather than leaving the query open-ended. The outputs are restricted to methods available in scikit-learn, ensuring that recommendations are directly usable in practice. All prompts are executed under zero-shot settings with deterministic inference, minimizing variance and making experiments repeatable. Once the data transformer pipeline is finalized, we proceed to model training using the hyperparameter (HP) combinations introduced in Section 4.1. These verified HP configurations, together with the finalized preprocessing and feature engineering pipeline, form the foundation for evaluating model performance under the dual objectives of fairness and accuracy. Since the LLM only recommends preprocessing methods, we do not directly evaluate the accuracy of its outputs. The final model training and evaluation, including both accuracy and fairness, are performed by AutoSklearn. The LLM's role is advisory, guiding and narrowing the preprocessing options rather than determining predictive outcomes.

Table 2: Examples of feature types, LLM-recommended preprocessing, and expected outputs.

| Feature Type | Example Input | LLM-Recommended Preprocessing | Expected Output After Prompting |
|---|---|---|---|
| Numerical (continuous) | age, income | Scaling (MinMax, Standard), log transform | "Apply StandardScaler to normalize income" |
| Categorical (low cardinality) | gender, marital_status | One-hot encoding, ordinal encoding | "Use one-hot encoding for marital_status" |
| Categorical (high cardinality) | zip_code, occupation | Target encoding, hashing | "Apply target encoding for occupation" |
| Binary attributes | default, owns_house | Leave as-is or convert to 0/1 | "Keep binary encoding (0/1) for default" |
| Missing-value columns | salary, credit_score | Imputation (mean, median, mode) | "Impute missing salary with median" |
| Correlated features | height_cm, height_inch | Feature selection / drop redundancy | "Drop height_inch due to correlation with height_cm" |

### 4.3 Pruning Strategy 1. Keep Diverse Batches in HP Space

We designed a technique called *Skipconf* that prunes our HP search space based on an iterative exploration of the search space. The input to Skipconf is the entire HP search space, represented as a list of all batches from Section 4.1. Suppose the number of such batches is $M$. The output is a list of $N$ batches that are *representative* of all $M$ batches, where $N \leq M$.

Intuitively, some batches in an HP space may be similar to each other in terms of their contribution to ML model performance (e.g., accuracy). Hence, from an HPO perspective, it should suffice to assess an ML model only on batches that are different from each other, yet collectively representative of all the batches in the generated HP space. To find such diverse batches, we designed a guided exploration technique. First, we assign an identifier (ID) to each generated batch, starting from 1, 2, ... The order between two batches is determined based on how they are created. Suppose we have two HPs (A and B) for an ML model. The range of values for A is 10–30, while for B it is 0–5. During our batch creation process, we created the following batches: (10,0), (20,0), (30,0), ..., (10,1), ..., (30,5). Here, the first batch is numbered as 1, the second as 2, and so on.

We enumerate the Cartesian product in a consistent order where the values of one hyperparameter (e.g., A) are fully exhausted before incrementing the next (e.g., B). This ensures that consecutive batches typically differ in at most one hyperparameter, making adjacency a reasonable proxy for potential similarity. Moreover, Skipconf validates this intuition by explicitly comparing performance across consecutive batches before skipping.

---

**Algorithm 1** Focused Region Finding

1: **Input:** $\xi$ (evaluated HP batches), $\kappa$ (list of hyper-parameters), $\rho$ (percentile value)
2: **Output:** $\zeta$ (dictionary of focused hyperparameter ranges)
3: **if** $\kappa = \emptyset$ **then return** None
4: $\alpha \leftarrow \{\phi \mapsto \emptyset \mid \phi \in \kappa\}$
5: **for all** entry $\in \xi$ **do**
6:      **for all** $\phi \in \kappa$ **do**
7:          $\alpha[\phi] \leftarrow \alpha[\phi] \cup \{\text{entry}[\text{config}][\phi]\}$
8: $\zeta \leftarrow \{\}$
9: **for all** $(\phi, \nu) \in \alpha$ **do**
10:      **if** $\nu \neq \emptyset$ **then**
11:          $\gamma \leftarrow \Big(\text{percentile}(\nu, 100 - \rho), \text{percentile}(\nu, \rho)\Big)$
12:          $\lambda \leftarrow \{v \in \nu \mid \gamma[0] \leq v \leq \gamma[1]\}$
13:          **if** $\lambda \neq \emptyset$ **then**
14:             $\zeta[\phi] \leftarrow \big(\min(\lambda), \max(\lambda)\big)$
15:          **else**
16:             $\zeta[\phi] \leftarrow \big(\min(\nu), \max(\nu)\big)$
17:      **else**
18:          $\zeta[\phi] \leftarrow (\emptyset, \emptyset)$
     **return** $\zeta$

---

Intuitively, the closer one batch is to another in terms of ordering, the more similar they are likely to be. For such similar batches, we may assume that the ML model will exhibit comparable performance. If the performance differs significantly, however, we can consider the batches diverse enough from each other. In such cases, all these batches are retained for further evaluation, as they contribute unique information about the search space.

The *Skipconf* algorithm is designed to accelerate exploration of the hyperparameter search space by selectively skipping configurations that are likely to yield redundant results. Each configuration batch is evaluated to obtain its accuracy $a_i$. The algorithm maintains three control parameters: an accuracy threshold $\tau_a$, a batch size $B$, and a skip percentage $p$. The decision to skip builds on the intuition that consecutive batches with similar performance provide little additional information about the search space. To formalize this similarity condition, we compare the accuracy of the current batch with that of the most recently evaluated batch. If the difference is within a predefined threshold, we mark the batches as similar:

$$|a_i - a_{\text{last}}| \leq \tau_a \tag{1}$$

then the two batches are deemed similar. A counter $c$ tracks the number of consecutive batches that satisfy this similarity condition. When the counter reaches the batch size threshold ($c \geq B$), *Skipconf* infers that the search is in a plateau region with little accuracy variation. At this point, the algorithm skips ahead by the following number of configurations:

$$s = \min\left(\lfloor p \cdot M \rfloor, ; M - i - 1\right), \tag{2}$$

where $M$ is the total number of configurations and $i$ is the index of the current batch. After skipping, the counter is reset, and the exploration resumes. This strategy helps the search process avoid wasting resources on regions of the hyperparameter space that yield very similar results, while still preserving enough variety to identify configurations that perform well and remain fair. The process terminates when all $M$ batches have either been explored or skipped. The output is a reduced set of batches that capture the diverse configurations of the HP space. To evaluate the ML model under this strategy, we recommend using a performance metric appropriate for the given dataset. For example, in a balanced dataset, accuracy would suffice.

### 4.4 Pruning Strategy 2. Keep Focused Regions of Batches

We consider a batch to be *satisfactory* if an ML model trained on its HP configuration achieves a good trade-off between fairness and accuracy. However, not all batches retained by our Skipconf technique (Strategy 1 in Section 4.3) necessarily produce such a trade-off. Therefore, we first filter the retained batches by evaluating each one using the Fairea methodology Hort et al. (2021), as discussed in Section 3 and illustrated in Figure 2. The identified 'satisfactory' batches can be scattered throughout our HP search space. Intuitively, the distribution of these batches could exhibit characteristics like Figure 4. This means that most batches could

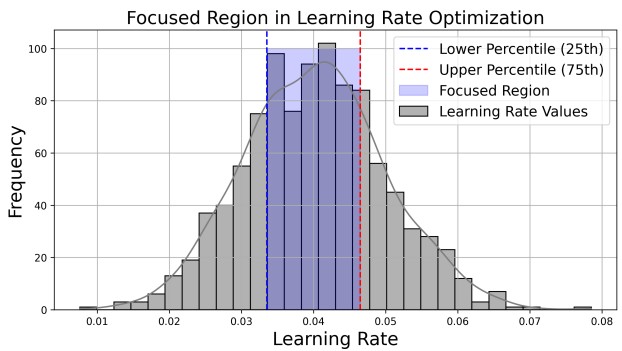

Figure 4: Concentration of batches in focused region

concentrate around an area, whereas the rest of the batches may form long tails on both ends. The concentration does not need to follow a normal distribution, but from an ML model perspective selecting HP configs from tails could lead to either overfitting or underfitting the model. As such, we further prune our list of 'satisfactory' batches from Strategy 2 by identifying the focused region among them. Algorithm 1 presents this approach. It takes as input the retained batches after the skipping process (i.e., the output from Section 4.3), along with the list of hyperparameters ($\theta$) and the percentile value ($\rho$). The output is the refined range of hyperparameters ($\delta$).

Let $D$ denote the set of all evaluated batches from Strategy 2. To isolate batches that provide a well-balanced performance in terms of accuracy and computational efficiency, we define specific regions $R$ (e.g., "win-win" and "good trade-off") and filter the batches accordingly:

$$D_{\text{filtered}} = \{\lambda \in D \mid \text{Region}(\lambda) \in R\} \tag{3}$$

Next, for each hyperparameter $\lambda_k$, we determine a focused range by computing percentile-based bounds over the filtered batches:

$$\text{Range}_{\lambda_k} = (\text{Percentile}_i(\lambda_k), \text{Percentile}_j(\lambda_k)) \tag{4}$$

Our choice of percentile-based filtering balances efficiency and interpretability. It systematically reduces the search space to the most promising central regions, thereby minimizing unnecessary evaluations while preserving strong performers in practice. The process begins by collecting the values of each hyperparameter across all batches in $D_{\text{filtered}}$. A focused range is then derived, capturing the central portion of values most likely to yield desirable performance. If the filtered values are sparse, the entire observed range is retained. Focused region identification is crucial in HPO because it refines the search by concentrating on the most promising hyperparameter batches after initial filtering. This approach prevents unnecessary evaluations of suboptimal candidates and ensures that both accuracy and fairness improvements are systematically

prioritized. By applying percentile-based selection and setting the cutoff at the top 25th percentile of the maximum accuracy among the candidates, we narrow the search space to batches offering the best trade-offs. Batches meeting or exceeding this cutoff are retained:

$$D_{\text{pruned}} = \Big\{ \lambda \in D_{\text{filtered}} \ \Big| \ \lambda_k \in \text{Range}_{\lambda_k} \text{ for all } k,$$
$$\text{Accuracy}(\lambda) \geq \text{Percentile}_{75}\big(\{\text{Accuracy}(\lambda') \mid \lambda' \in D_{\text{filtered}}\}\big)\Big\}. \tag{5}$$

Finally, the search space is refined by retaining only batches in which the hyperparameter values fall within the previously determined focused ranges. If no batches satisfy these conditions, the process terminates early, indicating that no suitable batches are available within the defined constraints.

### 4.5 Protection of Multiple Sensitive Attributes

In machine learning prediction tasks, datasets often contain more than one sensitive attribute that requires protection. For example, the Adult and COMPAS datasets include both sex and race, while the German Credit dataset involves sex and age. Considering these attributes jointly provides a more realistic and rigorous assessment of algorithmic fairness, revealing whether a mitigation approach generalizes beyond single-attribute protection. As an HPO approach, FAIRSPACE explicitly supports multiple and intersectional sensitive attributes, enabling fairness-aware search space pruning and evaluation across diverse subgroups.

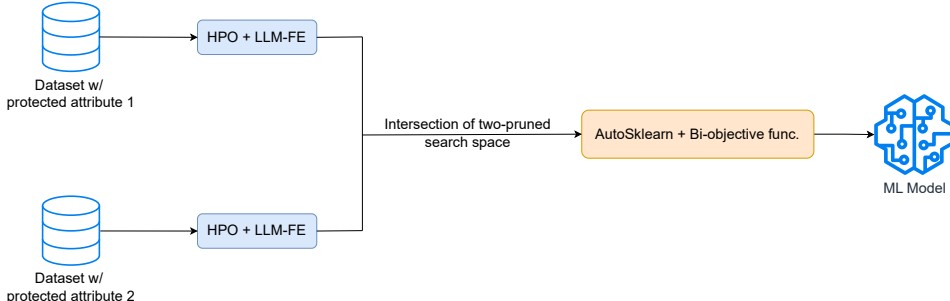

Figure 5: Workflow of FairSpace for multiple protected attributes.

To handle multiple protected attributes, we adopt a systematic strategy that evaluates each attribute individually before combining their outcomes as illustrated in Figure 5. For instance, in the Adult dataset, both race and sex are treated as protected attributes, and two separate training datasets are constructed. FAIRSPACE is applied to each dataset to obtain a pruned hyperparameter search space along with the most suitable feature preprocessing methods for that attribute. After obtaining these results, we compute the intersection of the pruned search spaces and feature engineering methods derived from both attributes. This intersected configuration is then used with Auto-Sklearn under a bi-objective optimization function. Such an approach provides greater flexibility in controlling fairness while ensuring that the machine learning model accounts for multiple sensitive dimensions simultaneously.

### 4.6 Fairness-Aware Model Training

We employ a dual-objective function that dynamically balances accuracy and fairness by adjusting their relative importance during training Nguyen et al. (2023). Specifically, the function uses a weighted-sum scalarization objective that integrates both accuracy and fairness metrics. In addition to hyperparameter configurations, the model incorporates dataset-specific feature engineering choices recommended by the LLM. Given a model $M$, a hyperparameter configuration $\mathbf{c}$, and a pruned dataset $D(z)$, the dual cost function is defined as:

$$\text{Cost}(M, \mathbf{c}, D(z)) = \beta \times f + (1 - \beta) \times (1 - a) \tag{6}$$

where $f$ denotes the fairness metric (e.g., SPD, EOD, AOD), $a$ the accuracy metric, and $\beta$ a dynamically adjusted parameter that controls the trade-off between the two.

## 5 Experimental Design

In this section, we outline our research questions and detail the experimental design used to evaluate FairSpace.

### 5.1 Research Questions (RQ)

We assess FairSpace by answering the following research questions.

- **RQ1. Fairness–Accuracy Trade-offs:** To what extent can FairSpace mitigate model bias without significantly compromising performance? We conduct a comprehensive comparison between FairSpace and existing bias mitigation methods across diverse decision-making scenarios.

- **RQ2. Effectiveness on Multiple Protected Attributes and Intersectional Fairness:** How well does FairSpace address fairness across multiple sensitive attributes and their intersections compared to existing methods? This question evaluates FairSpace's capability in handling individual attribute fairness as well as intersectional fairness, in comparison to existing methods.

- **RQ3. Effectiveness of FairSpace Strategies:** How effective are different fairness strategies within FairSpace? In particular, we investigate (i) the use of FairSpace with the default single-objective loss function of AutoSklearn and (ii) an ablation study on LLM-assisted feature engineering, to assess their impact on the fairness–accuracy trade-off.

- **RQ4. Efficiency of FairSpace:** How efficient is FairSpace with respect to computational overhead and time consumption? This RQ examines the additional time required by FairSpace over standard AutoML training pipelines, analyzing its suitability for real-world deployment.

### 5.2 Benchmark Datasets and Tasks

**Benchmark Datasets:** We evaluate FairSpace on five widely used datasets from recent fairness research: Adult Income Becker & Kohavi (1996), COMPAS ProPublica, German Credit Hofmann (1994), Bank Marketing Kaggle (2017), and MEPS-15 Agency for Healthcare Research and Quality (2015), covering key domains such as taxation, criminal justice, lending, banking, and public health. These datasets include sensitive attributes commonly linked to discrimination (sex, race, age). Our benchmark draws on three core areas: bias mitigation Chen et al. (2023), intersectional fairness Chen et al. (2024a), and fairness in AutoML Nguyen et al. (2023); Xiao et al. (2024). Unlike prior work that addresses isolated fairness aspects, we provide a unified evaluation framework with twelve tasks—nine single-attribute and three intersectional—capturing fairness across diverse and challenging settings characterized by class imbalance, noise, and varied feature scales as shown in Table 3.

**ML models:** Following the prior works Chen et al. (2022); Xiao et al. (2024); Chen et al. (2024a), we adopt four widely used machine learning models that are commonly employed to comprehensively evaluate the effectiveness of our proposed method: Logistic Regression (LR) Kleinbaum et al. (2002), Random Forest (RF) Breiman (2001), Support Vector Machine (SVM) Hearst et al. (1998), and Deep Neural Network (DNN) LeCun et al. (2015). These models are frequently used in real-world decision-making domains such as credit scoring, recidivism prediction, and income classification. DNNs are also included due to their widespread adoption in modern applications and increasing relevance in fairness-aware research. To ensure comparability and consistency, we follow the same model configurations and hyperparameter settings used in prior work Chen et al. (2022); Xiao et al. (2024); Chen et al. (2024a).

Table 3: Benchmark datasets and tasks.

| Task | Protected Attributes | Size | Favourable Labels | Label |
|---|---|---|---|---|
| 1. Adult-sex | Sex | 45,222 | 1 (income $> 50$k) | 0 (75.2%) |
| 2. Adult-race | Race | 45,222 | 1 (income $> 50$k) | 0 (75.2%) |
| 3. Compas-sex | Sex | 6,167 | 0 (no recidivism) | 0 (54.5%) |
| 4. Compas-race | Race | 6,167 | 0 (no recidivism) | 0 (54.5%) |
| 5. German-sex | Sex | 1,000 | 1 (good credit) | 1 (70.0%) |
| 6. German-age | Age | 1,000 | 1 (good credit) | 1 (70.0%) |
| 7. Bank-age | Age | 30,488 | 1 (subscriber) | 0 (87.3%) |
| 8. Mep-race | Race | 15,830 | 1 (utilizer) | 0 (82.8%) |
| 9. Adult-sex-race | Sex, Race | 45,222 | 1 (income $> 50$k) | 0 (75.2%) |
| 10. Compas-sex-race | Sex, Race | 6,167 | 0 (no recidivism) | 0 (54.5%) |
| 11. German-sex-age | Sex, Age | 1,000 | 1 (good credit) | 1 (70.0%) |

## 5.3 Baseline Methods

We compare FairSpace with a selected set of strong fairness-aware baselines that span different stages of the ML pipeline and support multiple protected attributes. These include Fair-SMOTE Chakraborty et al. (2021), a pre-processing method that balances the dataset using fairness-aware oversampling; MAAT Chen et al. (2022), which combines both pre- and post-processing techniques to optimize fairness-performance trade-offs; and FairMask Peng et al. (2022), a feature masking approach designed to mitigate bias through data transformation. We also include post-processing methods such as MirrorFair Xiao et al. (2024), which has shown strong performance on single-attribute fairness tasks, and FairHome Chen et al. (2024a), which is particularly effective in handling intersectional fairness involving multiple protected attributes. Additionally, we consider Fair-AutoML Nguyen et al. (2023), a fairness-aware extension of Auto-Sklearn that integrates bias mitigation directly into the AutoML pipeline and serves as a widely adopted, reproducible baseline.

Although several other fairness-aware methods exist, we limit our comparison to these six techniques due to their ability to handle multiple sensitive attributes. By evaluating FairSpace alongside these representative methods across various mitigation strategies, we offer a comprehensive and focused assessment of its effectiveness in balancing fairness and accuracy in multi-attribute settings.

## 5.4 Fairness Measures

We evaluate all methods using standard accuracy and fairness metrics, including group fairness measures—Statistical Parity Difference (SPD), Average Odds Difference (AOD), and Equal Opportunity Difference (EOD)—alongside six intersectional fairness metrics based on recent multi-attribute fairness literature Chen et al. (2022); Nguyen et al. (2023); Xiao et al. (2024); Chen et al. (2024a). To quantify the trade-off between fairness and accuracy, we use Fairea Hort et al. (2021), a state-of-the-art evaluation framework that computes the bias mitigation area based on selected performance and fairness metrics. The formulation of group and intersectional fairness metrics used in our evaluation is detailed in Section 3.

## 5.5 Experimental Design

Experiments were conducted using Python 3 on a machine with an 11-core CPU and 18 GB of unified memory. For each of the five datasets, we adopted a *StratifiedKFold* approach with 10 folds Awad & Fraihat (2023), ensuring that the class distributions were preserved in all divisions. Each experiment was independently repeated 20 times to obtain robust performance estimates, with the final results reported as the mean across runs, consistent with standard practice in fairness research Nguyen et al. (2023); Chen et al. (2022). Within each fold, the data was divided into 70% training and 30% test sets. The training portion was further split into 70% training and 30% validation data to facilitate hyperparameter tuning and model selection. FairSpace was executed first to prune and configure the search space, taking on average

Table 4: Accuracy and Fairness Measures. Lower fairness metric values indicate reduced bias. Each reported value is the average over 20 independent runs. Abbreviations: AS = AutoSklearn, FA = FairAutoML, FS = FairSMOTE, MT = MAAT, FM = FairMask, MF = MirrorFair, **FSP = FairSpace**. Sensitive attributes: S = Sex, A = Age, R = Race. Dataset: Adult = AD, Compas = CP, German = GM, Mep = MP and Bank = B. Due to space constraints, this table presents a summary; the complete results are available in the replication package Anonymous (2025).

| Task | Method | LR | | | | SVM | | | | RF | | | | DNN | | | |
|---|---|---|---|---|---|---|---|---|---|---|---|---|---|---|---|---|---|
| | | Acc | SPD | EOD | AOD | Acc | SPD | EOD | AOD | Acc | SPD | EOD | AOD | Acc | SPD | EOD | AOD |
| AD-S | AS | 0.84 | 0.17 | 0.08 | 0.07 | 0.85 | 0.17 | 0.07 | 0.07 | 0.86 | 0.18 | 0.06 | 0.05 | 0.85 | 0.18 | 0.04 | 0.05 |
| | FA | 0.80 | **0.02** | 0.009 | 0.02 | 0.82 | **0.03** | 0.05 | 0.06 | 0.85 | 0.16 | 0.04 | 0.06 | 0.80 | 0.02 | **0.0011** | 0.03 |
| | FS | 0.82 | 0.2 | 0.1 | 0.06 | 0.82 | 0.2 | 0.06 | 0.1 | 0.84 | 0.2 | 0.09 | 0.09 | 0.83 | 0.2 | 0.002 | 0.03 |
| | MT | 0.83 | 0.1 | 0.03 | 0.03 | 0.84 | 0.1 | 0.07 | 0.05 | 0.84 | 0.12 | 0.02 | 0.03 | 0.84 | 0.1 | 0.05 | 0.04 |
| | FM | 0.82 | 0.09 | 0.06 | 0.02 | 0.82 | 0.08 | 0.04 | 0.02 | 0.83 | 0.1 | 0.06 | **0.02** | 0.83 | 0.1 | 0.08 | 0.04 |
| | MF | 0.84 | 0.11 | 0.05 | 0.04 | 0.84 | 0.11 | 0.07 | 0.05 | 0.85 | 0.16 | 0.02 | 0.03 | 0.85 | 0.13 | 0.04 | 0.04 |
| | FSP | 0.82 | 0.03 | **0.007** | **0.006** | 0.83 | 0.04 | **0.04** | **0.02** | 0.85 | **0.1** | **0.02** | 0.04 | 0.82 | **0.02** | 0.02 | **0.01** |
| CP-R | AS | 0.67 | 0.19 | 0.25 | 0.18 | 0.69 | 0.18 | 0.25 | 0.14 | 0.69 | 0.22 | 0.20 | 0.16 | 0.68 | 0.13 | 0.17 | 0.16 |
| | FA | 0.62 | 0.08 | 0.001 | 0.14 | 0.59 | 0.05 | 0.14 | 0.1 | 0.66 | 0.14 | 0.14 | 0.13 | 0.64 | 0.13 | 0.07 | 0.04 |
| | FS | 0.66 | 0.31 | 0.22 | 0.25 | 0.64 | 0.28 | 0.24 | 0.26 | 0.67 | 0.12 | 0.02 | 0.07 | 0.62 | 0.23 | 0.07 | 0.08 |
| | MT | 0.65 | 0.06 | **0.008** | 0.05 | 0.64 | **0.04** | **0.007** | 0.03 | 0.64 | 0.08 | 0.02 | 0.06 | 0.64 | 0.05 | **0.02** | 0.04 |
| | FM | 0.68 | 0.17 | 0.14 | **0** | 0.64 | 0.18 | 0.15 | **0** | 0.65 | 0.14 | **0.01** | -0.05 | 0.68 | 0.18 | 0.13 | **-0.02** |
| | MF | 0.66 | 0.06 | 0.02 | 0.05 | 0.66 | 0.05 | 0.02 | 0.04 | 0.65 | 0.04 | 0.02 | 0.02 | 0.65 | 0.06 | 0.03 | 0.04 |
| | FSP | 0.66 | **0.03** | 0.01 | 0.05 | 0.67 | 0.05 | 0.02 | 0.04 | 0.66 | **0.008** | 0.03 | **0.007** | 0.65 | **0.01** | 0.04 | 0.05 |
| GM-A | AS | 0.72 | 0.06 | 0.27 | 0.06 | 0.71 | 0.08 | 0.14 | 0.05 | 0.75 | 0.1 | 0.1 | 0.06 | 0.72 | 0.12 | 0.17 | 0.05 |
| | FA | 0.70 | 0.06 | 0.08 | 0.09 | 0.70 | 0.06 | 0.08 | 0.09 | 0.73 | 0.07 | 0.08 | 0.12 | 0.71 | 0.17 | 0.08 | 0.09 |
| | FS | 0.73 | 0.27 | 0.13 | 0.22 | 0.73 | 0.25 | 0.2 | 0.21 | 0.79 | 0.14 | **0.02** | 0.05 | 0.74 | 0.13 | 0.06 | 0.05 |
| | MT | 0.71 | 0.07 | 0.007 | 0.02 | 0.75 | 0.05 | 0.007 | 0.03 | 0.77 | 0.12 | 0.06 | **0.03** | 0.74 | 0.07 | 0.06 | 0.09 |
| | FM | 0.74 | **0** | **0** | **0** | 0.71 | **0** | **0** | **0** | 0.69 | 0.27 | 0.12 | -0.15 | 0.72 | 0.24 | **0.03** | -0.2 |
| | MF | 0.74 | 0.05 | 0.05 | 0.07 | 0.75 | 0.05 | 0.05 | 0.1 | 0.76 | 0.05 | 0.04 | 0.07 | 0.74 | 0.08 | 0.06 | 0.08 |
| | FSP | 0.74 | 0.007 | 0.01 | 0.06 | 0.74 | 0.04 | 0.03 | 0.05 | 0.75 | **0.03** | 0.04 | 0.03 | 0.74 | **0.004** | 0.05 | **0.04** |
| B-A | AS | 0.90 | 0.08 | 0.05 | 0.09 | 0.90 | 0.08 | 0.16 | 0.07 | 0.91 | 0.13 | 0.12 | 0.07 | 0.90 | 0.1 | 0.13 | 0.06 |
| | FA | 0.89 | 0.03 | 0.02 | 0.06 | 0.89 | 0.02 | 0.01 | 0.04 | 0.90 | 0.07 | 0.16 | 0.07 | 0.90 | 0.04 | 0.09 | 0.08 |
| | FS | 0.89 | 0.32 | 0.54 | 0.37 | 0.89 | 0.29 | 0.5 | 0.33 | 0.90 | 0.22 | 0.23 | 0.17 | 0.89 | 0.24 | 0.25 | 0.19 |
| | MT | 0.90 | 0.1 | 0.06 | 0.05 | 0.90 | 0.12 | 0.09 | 0.07 | 0.91 | 0.13 | 0.06 | 0.05 | 0.90 | 0.13 | 0.06 | 0.05 |
| | FM | 0.76 | 0.12 | 0.1 | 0.08 | 0.77 | 0.1 | 0.15 | 0.16 | 0.78 | 0.05 | 0.06 | 0.15 | 0.79 | 0.11 | 0.14 | 0.1 |
| | MF | 0.90 | 0.05 | 0.04 | 0.03 | 0.90 | 0.05 | 0.04 | **0.03** | 0.90 | 0.06 | **0.05** | 0.04 | 0.90 | 0.09 | 0.06 | 0.05 |
| | FSP | 0.90 | **0.03** | **0.0002** | **0.02** | 0.90 | **0.002** | **0.01** | 0.04 | 0.90 | **0.05** | 0.07 | **0.0003** | 0.90 | **0.04** | **0.05** | **0.02** |
| MP-R | AS | 0.86 | 0.08 | 0.08 | 0.07 | 0.86 | 0.07 | 0.04 | 0.03 | 0.86 | 0.08 | 0.05 | 0.05 | 0.86 | 0.1 | 0.02 | 0.05 |
| | FA | 0.84 | 0.02 | 0.03 | 0.01 | 0.84 | 0.1 | 0.01 | 0.006 | 0.85 | 0.06 | **0.001** | **0.007** | 0.85 | 0.05 | 0.02 | 0.04 |
| | FS | 0.86 | 0.16 | 0.34 | 0.2 | 0.86 | 0.14 | 0.28 | 0.17 | 0.86 | 0.06 | 0.04 | 0.03 | 0.84 | 0.09 | 0.09 | 0.07 |
| | MT | 0.86 | 0.06 | 0.07 | 0.04 | 0.87 | 0.06 | 0.06 | 0.04 | 0.86 | 0.06 | 0.06 | 0.04 | 0.84 | 0.07 | 0.04 | 0.03 |
| | FM | 0.87 | 0.03 | 0.03 | 0.02 | 0.85 | **0.01** | -0.04 | -0.02 | 0.87 | 0.04 | -0.06 | -0.03 | 0.87 | **0.03** | 0.06 | **0.02** |
| | MF | 0.86 | 0.08 | 0.07 | 0.05 | 0.86 | 0.07 | 0.05 | 0.03 | 0.86 | 0.06 | 0.02 | 0.02 | 0.86 | 0.08 | 0.06 | 0.05 |
| | FSP | 0.85 | **0.02** | **0.009** | **0.01** | 0.85 | 0.02 | **0.03** | **0.001** | 0.86 | **0.009** | 0.01 | 0.03 | 0.86 | 0.05 | **0.01** | 0.02 |

15 minutes per run, after which Auto-Sklearn was run for 30 minutes. While prior work Nguyen et al. (2023) allocated 60 minutes per run, we reduced the runtime to 30 minutes given that FairSpace prunes and simplifies the search space, making exploration more efficient. For the LLM-assisted feature engineering stage, we employed Gemma3:12B Kamath et al. (2025) to generate dataset-specific preprocessing and feature engineering recommendations. FairSpace itself uses a small set of configuration parameters (e.g., $\delta$, $\beta$, $\sigma$), which we fixed to values determined in preliminary trials and kept constant across datasets to avoid dataset-specific tuning; the chosen values are documented in the supplementary material for reproducibility.

# 6 Results

## 6.1 RQ1. Fairness-Accuracy Trade-off

This RQ examines how well FairSpace and other methods mitigate bias on a single sensitive attribute. We design two evaluation settings: one assessing FairSpace's impact on performance and fairness, and another comparing it with SOTA techniques. MirrorFair Xiao et al. (2024) serves as the SOTA baseline, as it outperforms most methods except Fair-AutoML Nguyen et al. (2023) and Auto-Sklearn Feurer et al. (2020). All methods are evaluated over 20 repetitions across diverse tasks and classifiers.

Table 5: Fairness-performance trade-off.

| Method | win | good | inverted | bad | lose |
|---|---|---|---|---|---|
| AutoSklearn | 48% | 9% | 32% | 4% | 7% |
| FairAutoML | 38% | 30% | 19% | 8% | 5% |
| FairSMOTE | 22% | 14% | 25% | 4% | 35% |
| MAAT | 34% | 44% | 15% | 3% | 4% |
| FairMask | 30% | 23% | 30% | 4% | 13% |
| MirrorFair | 60% | 22% | 15% | 3% | 0% |
| FairSpace | 63% | 26% | 11% | 0% | 0% |

Table 6: Trade-off for multiple protected attributes.

| Method | win | good | bad | inverted | lose |
|---|---|---|---|---|---|
| AutoSklearn | 20% | 25% | 22% | 15% | 18% |
| FairSMOTE | 6% | 19% | 49% | 9% | 17% |
| FairMask | 25% | 61% | 7% | 2% | 5% |
| MAAT | 21% | 62% | 10% | 2% | 5% |
| MirrorFair | 67% | 17% | 11% | 5% | 0% |
| FairHome | 18% | 73% | 6% | 1% | 2% |
| FairSpace | 63% | 31% | 6% | 0% | 0% |

### 6.1.1 Impact of FairSpace on Model Performance and Fairness.

Table 4 summarizes accuracy and fairness metrics (SPD, EOD, AOD) for Auto-Sklearn, Fair-AutoML, FairSMOTE, MAAT, FairMask, MirrorFair, and FairSpace across various tasks and classifiers, as detailed in Section 5.3. FairSpace maintains accuracy while significantly reducing bias in 40 out of 96 cases. In tasks like Bank-Age and Mep-Race, it improves all performance and fairness metrics across all classifiers. Performance trade-offs are rare compared to MirrorFair and Fair-AutoML, though minor drops are seen in COMPAS-Sex and German-Sex. Overall, FairSpace achieves the highest fairness improvement in 42% of tasks, outperforming FairAutoML (12%), MirrorFair (3%), and closely followed by FairMask (25%).

### 6.1.2 Superiority of FairSpace over Existing Bias-Mitigation Methods.

We evaluate the superiority of FairSpace from two perspectives: (1) its ability to achieve a favorable trade-off between fairness and accuracy (i.e., maintaining high performance while improving fairness), and (2) its capacity to yield substantial improvements in fairness metrics alone, regardless of potential performance trade-offs. This distinction allows us to assess both FairSpace's balanced effectiveness and its raw fairness-enhancement potential.

**Superiority in fairness-performance trade-offs:** We use the Fairea benchmark Hort et al. (2021), following recent studies Chen et al. (2023) and SOTA Nguyen et al. (2023); Chen et al. (2022); Xiao et al. (2024). Fairea categorizes outcomes into five types—"win-win," "good," "bad," "inverted," and "lose-lose"—to measure how often a method simultaneously improves fairness and performance, as discussed in Section 3. Figure 5 presents these distributions for each bias mitigation method, based on 672 evaluation cases.

Table 5 shows that FairSpace achieves the highest proportion of "win-win" outcomes (63%), surpassing MirrorFair (60%) and all other methods. It also leads in overall effectiveness, with 89% of its outcomes classified as "win-win" or "good," outperforming MirrorFair (82%), MAAT (78%), and FairAutoML (68%). Notably, FairSpace avoids any "bad" or "lose-lose" outcomes, indicating stable performance. It also reduces "inverted" cases (13%) compared to MirrorFair and FairMask, where fairness improves but performance drops. These results confirm FairSpace's superior and consistent fairness–performance trade-off across diverse scenarios, including those involving deep learning methods Sun et al. (2022); Xiao et al. (2024).

**Superiority in overall fairness and performance improvement:** Table 7 reports the proportion of pairwise cases where each method significantly improves fairness over the others, based on the Fairea evaluation. Diagonal entries are 0% by definition, as a method cannot outperform itself. Among all methods, FairSMOTE (FS) and FairMask (FM) achieve the strongest relative improvements. FairSMOTE shows notable gains over

Table 7: Proportions of cases where each method improves fairness compared with other methods.

| Method | AS | FA | FS | MAAT | FM | MF | FAS |
|---|---|---|---|---|---|---|---|
| AS | 0% | 12.0% | 36.8% | 14.4% | 22.4% | 5.6% | 8.8% |
| FA | 16.4% | 0% | 37.1% | 15.0% | 22.9% | 6.4% | 2.1% |
| FS | 28.2% | 12.8% | 0% | 10.3% | 43.6% | 5.1% | 0% |
| MAAT | 19.4% | 13.7% | 37.1% | 0% | 24.2% | 1.6% | 4% |
| FM | 17.5% | 14.6% | 48.5% | 13.6% | 0% | 2.9% | 2.9% |
| MF | 16.9% | 14.9% | 29.2% | 12.3% | 22.1% | 0% | 4.6% |
| FAS | 16.5% | 12.1% | 28.2% | 15.0% | 21.4% | 6.8% | 0% |

FairAutoML (37.1%) and FairMask (43.6%), while FairMask, in turn, improves substantially over FairSMOTE (48.5%) and Auto-Sklearn (17.5%). MAAT also demonstrates competitive gains, improving over FairSMOTE in 37.1% of cases.

FairSpace (FAS) delivers consistent and balanced performance, improving fairness across all other methods: 16.5% over Auto-Sklearn (AS), 12.1% over FairAutoML (FA), 28.2% over FairSMOTE (FS), 15.0% over MAAT, 21.4% over FairMask (FM), and 6.8% over MirrorFair (MF). While its advantage over MirrorFair is relatively modest, FairSpace still achieves nearly seven percent improvement, underscoring its robustness.

Taken together, these results suggest that while FS and FM dominate in certain pairwise comparisons, FairSpace provides a more stable improvement profile across a broader set of competitors.

**Significance of difference in fairness between different methods:** Beyond the prior evaluation conducted using the Fairea framework, we perform a statistical comparison between each pair of methods using the Mann–Whitney U-Test Chen et al. (2022); Mann & Whitney (1947), a non-parametric test commonly used to determine whether one distribution tends to yield higher values than another. Consistent with prior work Chen et al. (2022); Xiao et al. (2024), we consider a fairness improvement to be statistically significant when the $p$-value is less than 0.05. We evaluate this for all fairness metrics used in our study and report the proportion of tasks (datasets $\times$ metrics) where each method statistically outperforms another.

Table 8: Proportion of cases where one method significantly improves fairness compared to another.

| Method | AS | FA | FS | MAAT | FM | MF | FAS |
|---|---|---|---|---|---|---|---|
| AS | 0% | 0% | 60% | 0% | 0% | 0% | 0% |
| FA | 73% | 0% | 76% | 0% | 0% | 0% | 0% |
| FS | 0% | 0% | 0% | 0% | 0% | 0% | 0% |
| MAAT | 71% | 0% | 77% | 0% | 0% | 0% | 0% |
| FM | 74% | 0% | 80% | 58% | 0% | 59% | 0% |
| MF | 68% | 0% | 73% | 0% | 0% | 0% | 0% |
| FAS | 88% | 65% | 84% | 69% | 0% | 81% | 0% |

As shown in Table 8, FairSpace (FAS) significantly improves fairness compared to Auto-Sklearn (88%), FairAutoML (65%), FairSMOTE (84%), MAAT (69%), and MirrorFair (81%), while showing no significant improvement over FairMask (0%). Among the baselines, FairMask exhibits moderate improvements over Auto-Sklearn (74%), FairSMOTE (80%), MAAT (58%), and MirrorFair (59%), but not over FairAutoML (0%). FairAutoML also shows notable improvements over Auto-Sklearn (73%) and FairSMOTE (76%). All other comparisons mostly result in 0%, indicating no statistically significant difference. These results reinforce the superiority and consistency of FairSpace in enhancing fairness across diverse datasets and evaluation metrics, while also highlighting some competitive strengths of FairMask and FairAutoML against selected baselines.

> **RQ1.** FairSpace outperforms existing bias mitigation methods by achieving the best fairness–accuracy trade-off, with 63% "win-win" outcomes and 89% classified as "win-win" or "good," surpassing Mirror-Fair, MAAT, and FairAutoML. It also delivers stronger fairness improvements than Auto-Sklearn (88%), FairAutoML (65%), FairSMOTE (84%), MAAT (69%), and MirrorFair (81%), while remaining competitive with FairMask.

## 6.2 RQ2. Effectiveness on multiple PA

Some datasets include more than one sensitive attribute that needs to be protected. Among the existing methods we tested, only six existing method are designed to handle multiple sensitive attributes Chakraborty et al. (2021); Peng et al. (2022); Chen et al. (2022); Xiao et al. (2024), as mentioned in their original papers. To answer this RQ, we compared the FairSpace with other SOTA methods like FairHome Chen et al. (2024a) and the other existing methods. The goal was to evaluate how well they balance model performance and fairness when addressing bias across multiple sensitive attributes. We performed this comparison using the Adult, COMPAS, and German datasets, all of which include multiple protected attributes. These datasets were specifically chosen to evaluate FairSpace's effectiveness in multi-attribute and intersectional fairness scenarios.

Table 6 presents the trade-offs of seven bias mitigation methods for multiple protected attributes, evaluated using the Fairea framework. FairSpace demonstrates the most consistent performance, with 63% win-win outcomes, 31% good trade-offs, and no cases of "inverted" or "lose," underscoring its stability in balancing fairness and accuracy. Although MirrorFair achieves the highest win-win rate (67%), its reliability is weakened by 11% bad and 5% inverted results. FairHome, MAAT, and FairMask primarily contribute to the "good" region (61%–73%) but exhibit lower win-win rates. In contrast, FairSMOTE performs poorly, with only 6% win-win outcomes and the majority (over 65%) falling in unfavorable categories. These findings highlight FairSpace's effectiveness in navigating fairness–accuracy trade-offs through its evolutionary optimization strategy.

**RQ2.** FairSpace achieves 63% win–win and 31% good outcomes across multiple sensitive attributes, with no inverted or lose–lose cases. Although MirrorFair reaches a higher win–win rate (67%), it produces more unfavorable results. FairHome, MAAT, and FairMask yield mainly good but fewer win–win outcomes, while FairSMOTE performs the weakest. Overall, FairSpace remains the most stable and reliable for intersectional fairness.

### 6.3 RQ3. Effectiveness of FairSpace strategies

This RQ evaluates the effectiveness of FairSpace's strategies by examining whether the combination of pruning, fairness-aware optimization, and LLM-guided feature engineering improves fairness–accuracy trade-offs compared to existing approaches. It further analyzes how each component contributes to the framework and assesses FairSpace's consistency and reliability across diverse datasets and classifiers.

#### 6.3.1 How FairSpace performs with single-objective function.

In the FairSpace pipeline, we integrated the bi-objective optimization function from Fair-AutoML during the in-processing stage (see Equation 6). This cost function jointly optimizes fairness and accuracy when training ML models, after the search space has been pruned. Because FairSpace prunes candidates by considering both fairness and accuracy, an open question is whether pruning alone is sufficient to achieve fair outcomes. To investigate this, we introduced FairSpace II, a variant in which the bi-objective function is replaced with the single-objective optimization of AutoSklearn, thereby training models using only the accuracy-driven loss while still applying LLM-guided feature engineering. We then compared FairSpace and FairSpace II to evaluate whether explicitly embedding fairness into the optimization objective yields more favorable fairness-accuracy trade-offs.

As shown in Table 9, FairSpace achieves a higher proportion of win-win outcomes (63%) compared to FairSpace II (58%), while both methods produce the same share of good trade-offs (26%). FairSpace also slightly reduces the occurrence of inverted cases (11% vs. 12%), and neither method generates bad trade-offs or lose-lose outcomes. The superiority of FairSpace stems from its bi-objective loss function,

Table 9: Trade-off distribution of FairSpace & FairSpace-II.

| Method | win | good | inverted | bad | lose |
|---|---|---|---|---|---|
| FairSpace | 63% | 26% | 11% | 0% | 0% |
| FairSpace II | 58% | 26% | 12% | 0% | 0% |

which simultaneously optimizes for fairness and accuracy, enabling the search process to identify models that are not only performant but also less biased. In contrast, FairSpace II relies on a single-objective loss from AutoSklearn that prioritizes accuracy, which explains its tendency to sacrifice fairness in marginal cases. These results confirm that bi-objective optimization yields more consistent fairness-accuracy improvements by explicitly embedding fairness into the training objective.

**RQ3.1** FairSpace outperforms its single-objective variant, FairSpace II, achieving more win-win cases and fewer inverted ones. While both leverage pruning and LLM-guided feature engineering, explicitly optimizing for fairness alongside accuracy yields more consistent trade-offs.

#### 6.3.2 To what extent does LLM-assisted feature engineering improve fairness–accuracy trade-offs compared to random selection of feature engineering methods?

To address this RQ, we conduct an ablation study that isolates the contribution of the LLM-assisted feature engineering stage in our pipeline. Specifically, we compare the performance of models trained with feature engineering methods recommended by the LLM against a baseline where methods are chosen randomly as discussed in Section 4.2. The objective is to demonstrate whether dataset-specific, LLM-guided decisions provide consistent improvements in accuracy and fairness metrics over unguided, random choices.

As shown in Table 10, LLM-assisted feature engineering consistently achieves higher performance than random selection across all benchmark datasets. Accuracy increases are observed across settings (e.g., from 0.81 to 0.84 on *Adult-Sex* and from 0.88 to 0.90 on *Bank-Age*), indicating that LLM-guided choices lead to more reliable predictive outcomes. More importantly, fairness improvements are pronounced, with SPD reductions ranging from 0.03 to 0.06 (e.g., from 0.12 to 0.07 on *Adult-Sex* and from 0.18 to 0.12 on *COMPAS-*

Table 10: Ablation study: LLM-assisted vs. Random FE across datasets. Bold indicates better performance.

| Dataset | Accuracy | | SPD ($\downarrow$) | |
|---|---|---|---|---|
| | Random | LLM-Assisted | Random | LLM-Assisted |
| Adult-Sex | $0.81 \pm 0.02$ | $\mathbf{0.84 \pm 0.01}$ | $0.12 \pm 0.03$ | $\mathbf{0.07 \pm 0.02}$ |
| COMPAS-Race | $0.66 \pm 0.03$ | $\mathbf{0.68 \pm 0.02}$ | $0.18 \pm 0.04$ | $\mathbf{0.12 \pm 0.03}$ |
| German-Age | $0.69 \pm 0.02$ | $\mathbf{0.72 \pm 0.01}$ | $0.10 \pm 0.02$ | $\mathbf{0.06 \pm 0.02}$ |
| Bank-Age | $0.88 \pm 0.01$ | $\mathbf{0.90 \pm 0.01}$ | $0.06 \pm 0.02$ | $\mathbf{0.03 \pm 0.01}$ |
| MEPS-Race | $0.85 \pm 0.02$ | $\mathbf{0.87 \pm 0.01}$ | $0.08 \pm 0.02$ | $\mathbf{0.04 \pm 0.01}$ |

*Race*). These findings demonstrate that LLM-assisted feature engineering not only enhances predictive accuracy but also systematically reduces bias, thereby strengthening the fairness–accuracy trade-off within the AutoML pipeline.

> **RQ3.2** LLM-assisted feature engineering outperforms random selection across all datasets, yielding higher accuracy and lower SPD. These results show that dataset-specific, LLM-guided choices improve both predictive performance and fairness, leading to stronger fairness–accuracy trade-offs.

### 6.4 RQ4. Time Efficiency of FairSpace

In this RQ, we investigate the efficiency of FairSpace in terms of computational time and resource consumption, comparing it against the state-of-the-art fairness-aware AutoML method, Fair-AutoML.

The Figure 6 illustrates the impact of runtime on FairSpace's performance by plotting accuracy against fairness, where lower fairness values indicate reduced bias. The color of each data point represents runtime, ranging from 15 minutes (green) to 60 minutes (salmon), while circle sizes correspond to different fairness metrics: Statistical Parity Difference (largest), Equalized Odds Difference (medium), and Average Odds Difference (smallest). A shorter runtime (15 mins, green) achieves high accuracy but results in poorer fairness, whereas increasing the runtime to 30 mins (blue) provides a better balance between accuracy and fairness, suggesting effective optimization.

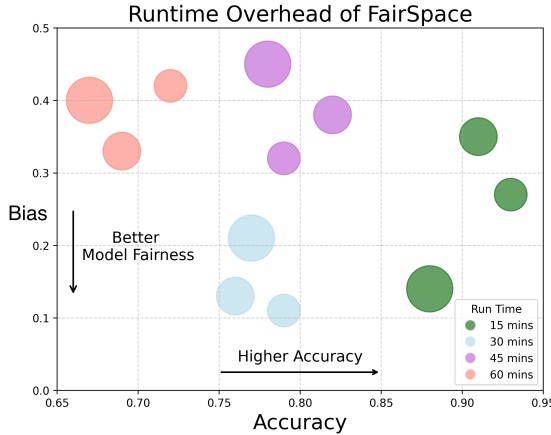

Figure 6: Runtime comparison of FairSpace, showing computational overhead and efficiency.

Beyond 30 minutes, fairness continues to improve slightly at 45 minutes (purple) and 60 minutes (salmon), but accuracy starts to degrade, indicating potential overfitting or diminishing returns in optimization. The 30-minute runtime emerges as the optimal choice, ensuring efficient training while maintaining fairness improvements, justifying its selection for other experiments.

> **RQ4.** FairSpace achieves optimal fairness and accuracy balance at a 30-minute runtime, maintaining efficiency without overfitting, while longer runtime reduce accuracy. This demonstrates FairSpace's time-effectiveness compared to methods needing extended durations, ensuring resource efficiency without sacrificing performance.

## 7 Discussion

To investigate the effectiveness of different pipeline configurations, we compare FairSpace, which combines pruning, LLM-assisted feature engineering, AutoSklearn, and a bi-objective fairness–accuracy function, with FairSpace III, which begins by initializing the search space, then trains and evaluates configurations, applies

LLM-assisted feature engineering, and finally assesses trade-offs using Fairea (i.e., FairSpace-III does not use the bi-objective optimization).

Table 11: Trade-off distribution of FairSpace and FairSpace-III.

| Method | win | good | inverted | bad | lose |
|--------|-----|------|----------|-----|------|
| FairSpace | 63% | 26% | 11% | 0% | 0% |
| FairSpace-III | 67% | 19% | 12% | 1% | 1% |

Table 11 presents the distribution of outcomes across the five Fairea regions: win, good, inverted, bad, and lose. As shown in Figure 7, the average runtime of FairSpace is 45 minutes compared to 15 minutes for FairSpace-III. In terms of fairness–accuracy outcomes, FairSpace achieves 63% win and 26% good cases, with no results falling into the bad or lose categories. FairSpace-III shows a slightly higher 67% win rate, though it introduces small proportions of bad (1%) and lose (1%) outcomes. These findings highlight that while FairSpace-III can marginally increase win cases, it does so at the expense of introducing failure scenarios. We conclude that FairSpace provides a more stable and robust fairness–accuracy balance, whereas FairSpace-III explores broader trade-offs with slightly higher risk.

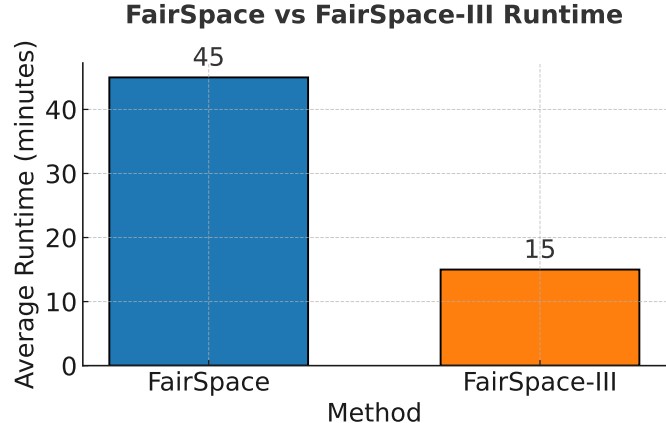

Figure 7: Comparison of runtime between FairSpace and FairSpace-III.

## 8 Related Works

**Bias Mitigation in ML.** Bias mitigation techniques in machine learning can be broadly categorized as pre-processing, in-processing, or post-processing Friedler et al. (2019); Hort et al. (2024). Pre-processing methods such as Disparate Impact Remover Feldman et al. (2015), Reweighing Kamiran & Calders (2012), Fair-SMOTE Chakraborty et al. (2021), Fairway Chakraborty et al. (2020), and FairMask Peng et al. (2022) modify training data or its representation to reduce bias before learning. In-processing approaches integrate fairness objectives directly into model training or optimization, as in MAAT Chen et al. (2022), Fax-AI Grabowicz et al. (2022), Equalized Odds Hardt et al. (2016), and Reject Option Classification Kamiran et al. (2012), or apply causality-driven adjustments and adaptive strategies in deep models Sun et al. (2022); Zhang & Sun (2022). Post-processing techniques such as MirrorFair Xiao et al. (2024) operate on model outputs, often leveraging ensembles or counterfactuals to improve fairness after training. Despite substantial progress, many existing methods either focus on specific fairness types, rely on manual or costly interventions, or risk sacrificing model accuracy for bias reduction.

**Intersectional Fairness.** Most prior studies focus on single protected attributes Chen et al. (2022); Peng et al. (2022); Chakraborty et al. (2021), yet intersectional bias—arising from the overlap of multiple identities—poses significant real-world harms. Buolamwini and Gebru Buolamwini & Gebru (2018) demonstrated this for gender and skin tone. Recent solutions, such as GRY Kearns et al. (2018), frame fairness as a two-player game for subgroup protection, while FairHOME Chen et al. (2024a) uses ensemble-based higher-order mutations to address complex intersectional bias. Empirical studies Wang et al. (2022) highlight the persistent challenges of achieving fairness across multiple attributes. Despite significant progress, existing methods often sacrifice accuracy for fairness or only address narrow bias types. Manual pre-processing and reliance on pre-defined search spaces limit broad adoption. Recent fairness-aware AutoML frameworks partly address these concerns.

**Fairness in AutoML.** Integrating fairness into Automated Machine Learning (AutoML) is a recent and growing area. Fair-AutoML Nguyen et al. (2023) incorporates fairness into the AutoML search process but relies on curated search spaces. Other work embeds fairness in hyperparameter optimization Wu & Wang (2021), but mainly targets group fairness and classification tasks. FALCC Lässig (2023) and Many-FairHPO Robertson et al. (2024) introduce local fairness objectives and human-in-the-loop processes, but at the expense of automation and scalability.

**LLM-Assisted Feature Engineering.** Recent studies have explored large language models (LLMs) for automating feature engineering in tabular data. Nam et al. introduce OCTree Nam et al. (2024), which uses decision tree–based reasoning as natural language feedback to refine feature generation. Ko et al. propose FeRG-LLM Ko et al. (2025), combining chain-of-thought dialogue and preference optimization to generate features and executable code. Abhyankar et al. present LLM-FE Abhyankar et al. (2025), framing feature engineering as an evolutionary program search guided by LLMs. These works demonstrate the potential of LLMs in feature discovery through reasoning and optimization, while challenges remain in prompt design, backbone selection, and fairness.

## 9 Threats to Validity

**Construct Validity.** To capture fairness accurately, we use established group and intersectional fairness metrics, including both worst-case and average-case variants, aligned with prior work Chen et al. (2022); Nguyen et al. (2023); Xiao et al. (2024); Chen et al. (2024a). While fairness is multifaceted, our study focuses on group-level metrics and excludes individual or causal fairness measures.

**Internal Validity.** We standardize data splits, seeds, preprocessing, and training across methods, with baselines covering multiple fairness stages and model types. Hyperparameters are informed by ablations and prior work, though some manual tuning remains. FairSpace depends on parameters ($\delta$, $b$, $\sigma$) fixed via prior work and empirical tuning, but their sensitivity is not yet systematically studied. To reduce bias, we release code, report all settings, and run repeated trials. Despite evaluating more configurations, FairSpace adds only modest runtime overhead by reusing run history and validation splits.

**External Validity.** We evaluate on five public datasets from diverse domains (justice, finance, health) and across multiple model types (LR, RF, SVM, DNN), supporting generalizability to fairness-critical tasks. However, our focus is on structured data; extending FairSpace to unstructured domains like images or text may require adapting genetic operations. Still, the core idea of fairness-guided search over run history remains broadly applicable.

## 10 Conclusion

We presented *FairSpace*, a fairness-aware AutoML. FairSpace combines LLM-assisted feature engineering with pruning strategies that reduce the search space while maintaining a strong balance between accuracy and fairness. Through systematic experiments across multiple datasets and models, we showed that FairSpace achieves comparable or better fairness–accuracy trade-offs while improving efficiency. Our pruning strategies—keeping diverse batches, skipping redundant regions, and focusing on central ranges—helped reduce the cost of hyperparameter optimization without losing high-quality configurations. The results highlight that FairSpace can make fairness-aware AutoML pipelines both effective and practical.

**Future Work.** While FairSpace effectively prunes the hyperparameter search space and improves efficiency, it has certain limitations. The focused-region filtering in Section 4.4 evaluates HPs independently using percentile-based bounds, which risks excluding extreme values that work only in combination with others. This limitation may cause missed configurations with favorable fairness–accuracy trade-offs. Future work should explore joint HP analysis through multivariate or correlation-aware pruning, as well as principled threshold selection. We also plan to extend FairSpace beyond classification to regression and generation tasks, broadening the scope of fairness-aware optimization.

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
