# OpenReview forum: "FairSpace: Search Space Pruning of AutoML for Fairness-Accuracy Trade-off"
_TMLR — Rejected by TMLR_

### Review · Reviewer_d2qz · 2026-02-06

**Summary Of Contributions:**

Ensuring fairness when there are multiple sensitive attributes often requires computationally heavy hyperparameter tuning and feature engineering in AutoML systems. The paper proposes FAIRSPACE framework which (i) performs hyperparameter tuning and (ii) feature engineering (LLM assisted).
Experiments (five datasets) show better fairness-utility tradeoffs with lower computational costs (against SOTA, six baselines).

The problem of intersectional bias mitigation is indeed challenging, which is the focus of the paper in the context of AutoML frameworks.

The proposal involves two stages: (i) LLM-assisted feature engineering + preprocessing strategies. (ii) Hyperparameter space pruning using guided exploration strategy. Finally, use the refined space for model training using bi-objective function targeting fairness and accuracy.

The proposed strategy seems heuristics which is experimented only on five datasets and four models considered. Also experiments include only one LLM model. Only section 6.2 presents experiments dealing with multiple protective attributes. Overall, the problem context is timely, but the paper doesn't provide enough empirical evidence in support of there proposed solution.

**Audience:**

Yes

**Audience Explanation:**

The problem context is timely and appealing to many fairness researchers. But, the paper's results are not convincing at the moment.

**Claims And Evidence:**

No

**Claims Explanation:**

The LLM assisted strategy for scoping the hyperparameter space as well as preprocessing steps seems heuristically motivated. I couldn't find discussion on why the recommended strategies are optimal or best in any way or reliable in general? Is it because the datasets studied are well-explored and strategies are aware to chosen LLM model? Also, only one LLM model is studied (Gemma3). Would the results change with different LLMs? Why Gemma3 was chosen?

In Section 4.5, as the paper motivates with challenges in multiple protected attributes settings, what happens if we have many sensitive attributes leading to intersection of pruned-spaces essentially empty?

Figure 6 is quite confusing. For higher runtimes, the fairness actually degrades as well as accuracy. So, as a practitioner, how should one decide when to stop? In section 6.4, this discrepancy is attributed to overfitting. Is this also a reason for other baseline methods for poor performance since run for longer times?

There is no formal argument that the proposed strategies will lead to optimal tradeoffs.

**Requested Changes:**

1. The term "FAIRSPACE" and "FairSpace" can be made consistent throughout the paper.

2. What are 'batches' mentioned at the start of in Section 4.3 (It refers to Section 4.1, but there is no mention of batches!). I assume it is hyperparameter configurations.

3.  In Table 9, FairSpace II proportions don't add up to 100%.Could you please explain?

4. Although the paper is motivated by the challenge of intersectional fairness, the experimental section is mostly about single protective attribute settings. Only Section 6.2 presents related to intersectional fairness. Does Table 6 provide average values over all three datasets?

5. In section 5.2, it says there are twelve tasks, but the Table 3 lists only 11.

---

> ### Author Response · Authors · 2026-02-23
> **Official Comment by Authors**
>
> We sincerely thank the reviewer for the thoughtful and detailed feedback. Below we clarify the scope of our work and address the concerns raised.
>
>
> **Question 1: Heuristic nature of LLM-assisted strategy**
>
> The LLM-assisted strategy is heuristic by design. We do not claim it produces optimal preprocessing strategies. Its role is to narrow the search space in a dataset-aware manner, not to determine the final model. Fairness and accuracy outcomes are determined by the downstream AutoML optimization stage.
>
> The LLM receives only structured meta-features (e.g., imbalance, group distribution) and selects from a predefined set of scikit-learn transformations. It is not given prior knowledge about specific benchmark datasets beyond these summaries.
>
> We chose Gemma3 because it is a strong open-source model that supports reproducible and deterministic inference, which is important for experimental consistency. Different LLMs may yield slightly different preprocessing suggestions, but since recommendations are constrained and all retained configurations are evaluated by AutoML, moderate variation is unlikely to fundamentally alter conclusions. We will clarify that exploring multiple LLMs is future work.
>
>
> **Question 2: Many sensitive attributes and empty intersections**
>
> We prune the search space separately per protected attribute and then take the intersection. With many sensitive attributes, the intersection could theoretically become very small or empty.
>
> However, pruning is conservative: we retain configurations in win-win and good trade-off regions rather than applying strict thresholds, preserving diversity. If the focused region becomes too restrictive, the algorithm can revert to a broader filtered set instead of enforcing an empty intersection.
>
> We acknowledge that with many attributes, adaptive relaxation of pruning thresholds may be required and will clarify this limitation.
>
>
> **Question 3: Runtime degradation in Figure 6**
>
> Figure 6 shows that beyond a certain runtime, both fairness and accuracy may degrade. This is typical in AutoML systems due to overfitting to the validation set, reducing test generalization.
>
> In practice, runtime should be treated as a computational budget parameter. We recommend selecting it based on validation trends: when performance plateaus or declines, further search is unlikely to help.
>
> For baselines, we followed runtime settings from prior work to ensure fair comparison. Longer runtimes may also contribute to overfitting in some baselines. FairSpace benefits from pruning, which reduces unnecessary exploration, making extended runtime less beneficial.
>
> We will revise the discussion to provide clearer practical guidance.
>
>
> **Question 4: No formal optimality guarantees**
>
> We agree that FairSpace does not provide formal guarantees of optimal fairness–accuracy trade-offs. The pruning and LLM-assisted strategies are heuristic and designed for efficient exploration in large search spaces where exhaustive search is infeasible.
>
> FairSpace narrows the space and applies explicit bi-objective optimization. Its effectiveness is demonstrated empirically, but we do not claim theoretical optimality. We will clarify this explicitly.
>
>
>
> **Question 5: FAIRSPACE vs FairSpace**
>
> We agree and will standardize the spelling to “FairSpace” throughout.
>
>
> **Question 6: Meaning of “batches”**
>
> A “batch” refers to a hyperparameter configuration (or grouped unit) defined in Section 4.1 (“Initialize Search Space as Batches of HP Configurations”). Skipconf evaluates these batches and uses parameters such as batch size B when skipping plateaus.
>
> **Question 7: Table 9 proportions**
>
> This was a typo. The corrected table is:
>
> ## Trade-off Distribution: FairSpace vs FairSpace-II
>
> | Method       | Win-Win | Good Trade-Off | Inverted | Bad Trade-Off | Lose-Lose |
> | ------------ | ------- | -------------- | -------- | ------------- | --------- |
> | FairSpace    | 63%     | 26%            | 11%      | 0%            | 0%        |
> | FairSpace II | 58%     | 26%            | 12%      | 4%            | 0%        |
>
>
> **Question 8: Intersectional fairness experiments**
>
> RQ2 uses Adult, COMPAS, and German with multiple protected attributes, and Table 6 summarizes trade-off distributions over these multi-attribute settings. We acknowledge that most experiments focus on single-attribute settings. In the revision, we will add statistical tests to strengthen the intersectional fairness analysis.
>
> **Question 9: Twelve vs eleven tasks**
>
> This is a typo. Section 5.2 states “twelve tasks,” but Table 3 lists 11. We will correct the text to “eleven tasks.”
>
> Thank you again for the constructive feedback.

---

### Review · Reviewer_B7YJ · 2026-02-08

**Summary Of Contributions:**

The paper proposes FairSpace, a meta-heuristic for optimal hyper-parameter selection for designing human-centric decision-makers that should be both accurate and fair. The problem is particularly difficult when the fairness objective involves *multiple* protected attributes of the population, like being fair based on both gender and race. Since each protected attribute typically requires a different strategy for hyper-parameter optimizations and dataset pre-processing, the overall configuration space of the parameter search problem becomes intractable. FairSpace involves a combination of LLM-guided pruning of the hyper-parameter space (via automatically generated prompts), as well as accelerated search over the parameter space that discards parameter regions that are suspected to not produce good learning accuracy (because an adjacent region resulted in bad accuracy). Experiments have been performed on a number of popular data sets from the literature, and the effectiveness has been compared against state-of-the-art approaches.

**Audience:**

Yes

**Audience Explanation:**

As learned agents are being increasingly used in human-centric decision-making tasks, the fairness of these agents is becoming an important design criterion. This has led to a high volume of works on fairness in machine learning, and the manuscript under review falls in that category.

**Claims And Evidence:**

No

**Claims Explanation:**

Firstly, it has been argued that the presented approach "reduces computational overhead while adapting the pipeline more effectively to the dataset" (first paragraph, Section 4.2). I think this claim is misleading. The tremendous complexity of querying LLMs is totally ignored, let alone the complexity that was involved in training those LLMs.

Secondly, from the numbers shown in the experiments, it is unclear how much we gain when compared against the existing methods. In Section 7, the authors essentially dismissed their own alternative called FairSpace-III because it has "marginal" improvement over FairSpace, based on the numbers in Table 11. If I now use the same scale (63% versus 67% win rate, etc.), the gains of FairSpace over the existing approaches (as presented in Section 6) become negligible.

Thirdly, if LLM is an integral part of the tool, I would like to see some comparisons across different LLMs. It is widely accepted that different LLMs perform differently on different kinds of tasks, and most are known to be notoriously bad in logical reasoning tasks. Here the LLM is used to solve some logical task, and I am skeptical how well this works in general. In Section 6.3.2, the authors compare the LLM-based pruning with a random pruning, which I do not find convincing. Is it that surprising that LLMs are at least as good as a randomized solver?

Finally, it has been rightly pointed out that FairSpace itself uses some hyper-parameters (last paragraph of Section 5), but no ablation studies have been presented regarding their effect on the output. It was argued that their values are fixed as per preliminary trials, and were kept constant to avoid dataset-specific tuning. But this does not prove that the chosen parameters will work for benchmarks not considered in the paper. The point of ablation studies is to give the audience a feeling of how the hyper-parameters affect performance, so that they can be tuned if needed. This is a crucial component that is totally missing.

**Requested Changes:**

~ Critical concerns ~

1) The part with LLMs must be revisited, see my comments on whether "claims made (are) supported by accurate, convincing and clear evidence". Other than what I said already, I have some further thoughts on the very use of LLMs. Right now, the LLM component is provided structured inputs with a list of available meta features and a clear goal. Furthermore, the LLM is restricted to run in the deterministic inference mode. Under these very restrictive constraints, I wonder why LLMs are the appropriate choice. For instance, why not use a handcrafted decision-tree that would provide the required guidance?

2) An ablation study with respect to FairSpace's hyper-parameters ($\delta$, $\beta$, $\tau_a$, $p$, etc.) is needed. In Section 4.4, only the top 25th percentile of configurations has been selected. To me even this is a hyper-parameter. Then one of the things that is particularly problematic is the 30-minute runtime. In Section 5.5, the 30-minute runtime appeared like a disadvantage over the state-of-the-art. But then, in Section 6.4, it became clear that if the program is run too long, the performance could degrade due to overfitting, and the 30-minute cutoff time turned out to be the optimal choice. In light of this, I find the earlier pitch unfair. I recommend the runtime to be also treated like a hyper-parameter.

~ Non-critical, comments for improvement ~

3) Section 4.5: A compositional approach has been proposed, where the individual requirements for two separate fairness objectives (specified using the sensitive attribute) are composed to obtain the overall requirements for the combined fairness objective. I would imagine this to work when the two attributes are uncorrelated. Can you provide some intuition on why this approach is reasonable? I am thinking whether it could happen that some hyper-parameter P is required when both attribute 1 and 2 are present, but P is not required if any one of them is present. Is it guaranteed that such a situation will never occur?

4) It would be good to comment on problems related to the intersection of group and individual fairness, which fits with the high-level story. The combination of group and individual fairness is one of the emerging classes of fairness specifications. Would it be easy to extend the proposed approach to this setting?

---

> ### Author Response · Authors · 2026-02-23
> **Official Comment by Authors**
>
> We sincerely thank the reviewer for the thoughtful feedback and detailed review.
>
> **Question 1: Why use an LLM instead of a handcrafted decision tree?**
>
> Our use of an LLM is motivated by flexibility and scalability. The mapping from dataset meta-features to suitable preprocessing strategies is complex and highly context-dependent. Interactions among class imbalance, group representation, missing values, and feature correlations can produce different preprocessing needs depending on their combination. Encoding all such interactions in a handcrafted decision tree would require extensive domain-specific rules and continual updates as new datasets or preprocessing options are introduced.
>
> The LLM serves as a meta-level reasoning module that adapts to varying meta-feature combinations without manual rule engineering. Importantly, it operates under strict constraints: it receives structured inputs, selects only from a predefined set of transformations, and runs deterministically. It does not influence model training directly but only narrows the search space.
>
> We acknowledge that a rule-based alternative could be explored. We will clarify that the LLM is a flexible and extensible design choice rather than the only possible implementation, and discuss rule-based systems as future work.
>
> **Question 2: Hyper-parameters, percentile threshold, and runtime**
>
> Runtime is already treated as a computational budget parameter. In Figure 6, we analyze performance across different runtime budgets and show how fairness and accuracy evolve over time. The 30-minute budget was selected empirically: longer runtimes did not improve results and sometimes led to degradation due to overfitting in the AutoML search. Our intention was not to frame 30 minutes as a disadvantage, but to show that FairSpace achieves competitive or superior results with a smaller budget than prior work using 60 minutes. We agree runtime should be interpreted as a tunable budget and will clarify this wording.
>
> Regarding δ, β, τ, and the 25th percentile threshold: these were fixed across all datasets to avoid dataset-specific tuning and ensure fair comparison. While they can be viewed as hyper-parameters, our goal was to demonstrate robustness without per-dataset adjustment. We will make this design choice clearer and emphasize the existing runtime sensitivity analysis.
>
> **Question 3: Compositional approach for multiple sensitive attributes**
>
> Our compositional strategy does not assume uncorrelated attributes. We prune the search space separately for each sensitive attribute, then take the intersection and evaluate configurations using intersectional fairness metrics (worst-case and average-case variants).
>
> The intuition is that configurations that perform well for each attribute individually are more likely to generalize to their intersection than those unfair for at least one attribute. Final selection is based explicitly on intersectional fairness metrics, so any configuration failing under the combined objective is filtered out during bi-objective optimization.
>
> We agree it is theoretically possible that a hyper-parameter P is required only when both attributes are present but not when considered individually. Our method does not provide a formal guarantee against such cases. It is heuristic: the intersection narrows the space to robust configurations, and joint evaluation explicitly enforces intersectional fairness. We will clarify this limitation and emphasize that the approach is not guaranteed optimal.
>
> **Question 4: Group and individual fairness**
>
> FairSpace focuses on group and intersectional fairness using SPD, EOD, AOD, and their intersectional variants. We do not model individual fairness, which requires defining similarity between individuals and enforcing consistency constraints.
>
> Conceptually, FairSpace could be extended: the bi-objective optimization could incorporate an individual fairness term alongside group fairness and accuracy. However, this would require defining an appropriate similarity metric, computing pairwise or local constraints, and managing increased computational cost.
>
> Thus, while the architecture could support such an extension in principle, it would require careful formulation and efficient implementation. We will highlight this as an important direction for future work.
>
> Thank you again for the constructive feedback.

---

### Review · Reviewer_yX6b · 2026-02-09

**Summary Of Contributions:**

The paper introduces FairSpace, a fairness-aware Automated Machine Learning (AutoML) framework that jointly optimizes hyperparameter optimization (HPO) and feature engineering (FE) to balance predictive accuracy with bias mitigation. A primary contribution is the integration of Large Language Models (LLMs) to analyze dataset meta-features and recommend context-aware preprocessing strategies, which, alongside novel search space pruning techniques (like Skipconf) and focused region analysis, reduces computation time by eliminating redundant configurations.

**Audience:**

Yes

**Audience Explanation:**

The paper addresses challenges in responsible AI and AutoML using novel methodologies relevant to modern ML research. It handles the complex challenge of intersectional fairness across multiple protected attributes with a novel pruning strategy. Hence, it directly targets TMLR individuals whose interest is in scalable, responsible machine learning.

**Broader Impact Concerns:**

Addressing the following could be helpful.

- The risk that the LLM itself might hallucinate or reflect inherent biases from its training data.
- Is there a possibility that it may claim a model is "fair" as a group while it still discriminates against specific individuals within those groups?

**Claims And Evidence:**

Yes

**Claims Explanation:**

The authors validate their claims through extensive experiments on five datasets against six baselines, showing FairSpace achieves "win–win" outcomes in 63% of cases, surpassing state-of-the-art methods like MirrorFair and FairAutoML. Statistical tests confirm significant fairness improvements over these baselines, while runtime analysis proves a ~25% reduction in computation time. Additionally, ablation studies explicitly demonstrate that the LLM-assisted feature engineering and bi-objective optimization are key drivers of these results.

**Requested Changes:**

There are many notations that are used in the algorithm whose definitions are missing. For example, what does $\alpha$ signify? What are $\phi,\nu)$ in $\alpha$? An overview of the algorithm can resolve this.

The equations are not clearly written. It is difficult for a reader to understand the equations with undefined variables and functions. Some examples are as follows.
1) Eqn 2: What is $p$, what does ",;" mean in your min() function?
2) $\mathrm{Region}(\lambda)$ formal definition missing while being used in eqn 3

---

> ### Author Response · Authors · 2026-02-23
> **Official Comment by Authors**
>
> Thank you for your detailed feedback and thoughtful suggestions. Below we provide concise responses to the raised concerns.
>
> **Question 1: Undefined notations in Algorithm 1**
>
> Thank you for pointing this out. We agree that some notations were not sufficiently defined.
>
> * **α** denotes a temporary dictionary mapping each hyperparameter ϕ ∈ K to the set of its observed values across the filtered batch set ξ.
> * **ϕ (phi)** represents an individual hyperparameter in the hyperparameter set K.
> * **ν (nu)** denotes the collection of values corresponding to hyperparameter ϕ extracted from all configurations in ξ.
>
> Thus, α[ϕ] stores all observed values of hyperparameter ϕ among satisfactory batches, which are then used to compute percentile-based focused ranges.
>
> In the revision, we will:
>
> * Add a short symbol table before Algorithm 1.
> * Provide a high-level overview:
>
>   * Step 1: collect hyperparameter values from satisfactory batches.
>   * Step 2: compute percentile bounds.
>   * Step 3: derive focused ranges.
> * Clarify variable roles in the algorithm caption.
>
>
> **Question 2: Unclear equations**
>
> **(a) Equation (2)**
> [
> s = \min(\lfloor p \cdot M \rfloor, M - i - 1)
> ]
>
> Definitions:
>
> * **p**: skip percentage, a user-defined fraction (p ∈ (0,1)).
> * **M**: total number of configurations.
> * **i**: index of the current configuration.
> * ⌊·⌋: floor operator.
>
> The expression ensures that:
>
> 1. At most a fraction p of configurations is skipped.
> 2. The algorithm never skips beyond the remaining search space.
>
> We will replace the semicolon with a comma in the min function, define variables immediately after the equation, and add a short explanatory sentence for clarity.
>
> **(b) Equation (3)**
> [
> D_{\text{filtered}} = {\lambda \in D \mid \text{Region}(\lambda) \in R}
> ]
>
> * **λ** denotes a hyperparameter configuration.
> * **Region(λ)** maps λ to one of five Fairea trade-off regions: win–win, good trade-off, bad trade-off, inverted, lose–lose.
>
> Let:
>
> * a(λ): accuracy of λ
> * f(λ): fairness metric (lower is better)
> * a_b, f_b: baseline accuracy and fairness
>
> Regions are defined as:
>
> * **win–win**: a(λ) > a_b and f(λ) < f_b
> * **good trade-off**: fairness improves with acceptable accuracy drop
> * **inverted**: accuracy improves but fairness worsens
> * **bad trade-off**: both deteriorate beyond acceptable boundary
> * **lose–lose**: a(λ) < a_b and f(λ) > f_b
>
> R specifies retained regions (in our experiments: {win–win, good trade-off}).
>
> We will provide formal definitions before Equation (3) and clarify λ and the Fairea boundaries.
>
>
> **Question 3: LLM hallucination or bias risk**
>
> We acknowledge that LLMs may hallucinate or reflect training biases. However, in our framework, the LLM plays a constrained role:
>
> * It does not generate predictions or influence training directly.
> * It receives structured meta-features (class imbalance, group distribution, missing values).
> * It selects preprocessing methods from a predefined set of scikit-learn transformations.
> * Deterministic prompting reduces variability.
>
> Final fairness and accuracy results are determined solely by the AutoML process. We will clarify this design choice and explicitly discuss LLM limitations in the revision.
>
>
> **Question 4: Group vs. individual fairness**
>
> Our work focuses exclusively on group fairness. We evaluate SPD, EOD, and AOD, including worst-case and average-case intersectional variants, consistent with prior literature.
>
> We acknowledge that satisfying group-level metrics does not guarantee individual fairness. Addressing individual fairness requires a different formulation and is beyond the scope of this study. We will clarify this limitation explicitly.
>
>
> We appreciate the constructive feedback and will incorporate these clarifications to improve readability and rigor.

---

### Decision · Action_Editor_2ejz · 2026-06-01

**Recommendation:** Reject

**Audience:**

Yes

**Audience Explanation:**

The problem studied is indeed very interesting as ensuring fairness as well as accuracy is one of the most interesting topics. Therefore, if the proposed techniques were adequately supported, the paper would have attracted a large audience.

**Claims And Evidence:**

No

**Claims Explanation:**

The empirical evaluation does not adequetly support the claim as the evaluation is very limited to one specific strategy of the usage of LLM models and also based on choices of runtime as well as ignoring other baselines that have not been fully justified. Reviewers have pointed out these concerns and on the balance, I think the paper requires significant revision.

**Resubmission Of Major Revision:**

The authors may consider submitting a major revision at a later time.